# Usp22 Deficiency Leads to Downregulation of PD-L1 and Pathological Activation of CD8^+^ T Cells and Causes Immunopathology in Response to Acute LCMV Infection

**DOI:** 10.3390/vaccines11101563

**Published:** 2023-10-05

**Authors:** Justa Friebus-Kardash, Theresa Charlotte Christ, Nikolaus Dietlein, Abdelrahman Elwy, Hossam Abdelrahman, Lisa Holnsteiner, Zhongwen Hu, Hans-Reimer Rodewald, Karl Sebastian Lang

**Affiliations:** 1Institute of Immunology, Medical Faculty, University of Duisburg-Essen, 45147 Essen, Germany; theresacharlotte.christ@uk-essen.de (T.C.C.); abdelrahman.elwy@uk-essen.de (A.E.); hossam.abdelrahman@uk-essen.de (H.A.); lisa.holnsteiner@uk-essen.de (L.H.); zhongwen.hu@uk-essen.de (Z.H.); karlsebastian.lang@uk-essen.de (K.S.L.); 2Department of Nephrology, University Hospital Essen, University Duisburg-Essen, 45147 Essen, Germany; 3Division of Cellular Immunology, German Cancer Research Center, 69120 Heidelberg, Germany; n.dietlein@dkfz-heidelberg.de (N.D.);

**Keywords:** ubiquitin-specific peptidase 22 (Usp22), LCMV, PD-L1 downregulation, activation of CD8^+^ T cells, liver failure

## Abstract

Ubiquitin-specific peptidase 22 (Usp22) cleaves ubiquitin moieties from numerous proteins, including histone H2B and transcription factors. Recently, it was reported that Usp22 acts as a negative regulator of interferon-dependent responses. In the current study, we investigated the role of Usp22 deficiency in acute viral infection with lymphocytic choriomeningitis virus (LCMV). We found that the lack of Usp22 on bone marrow-derived cells (Usp22^fl/fl^ Vav1-Cre mice) reduced the induction of type I and II interferons. A limited type I interferon response did not influence virus replication. However, restricted expression of PD-L1 led to increased frequencies of functional virus-specific CD8^+^ T cells and rapid death of Usp22-deficient mice. CD8^+^ T cell depletion experiments revealed that accelerated CD8^+^ T cells were responsible for enhanced lethality in Usp22 deficient mice. In conclusion, we found that the lack of Usp22 generated a pathological CD8^+^ T cell response, which gave rise to severe disease in mice.

## 1. Introduction

A body of evidence suggests the intersection of viruses and autoimmunity. Several previous works postulated that viruses such as cytomegalovirus virus, Epstein–Barr virus, hepatitis B and C, and SARS-CoV-2 have a high potential to trigger autoimmune processes in genetically predisposed subjects [1,2,3]. An inappropriate immune response is the cardinal feature of COVID-19 infection; it leads to a severe course of viral infection and also promotes the development of a number of autoimmune diseases [3]. In a subset of COVID-19 patients with severe COVID-19 pneumonia, a robust production of type I interferon tracked organ damage [3]. The appearance of functional autoantibodies after COVID-19 infection was discussed to contribute to virus-mediated autoimmunity [3]. Chronic hepatitis C progresses to cirrhosis of the liver, end-stage liver failure, or hepatocellular carcinoma because of the persistent activity of cytotoxic CD8^+^ T cells, which initiate chronic inflammatory processes in the liver and can also induce autoimmune phenomena. However, the mechanisms by which viral infection can tip the host’s immune response toward the loss of tolerance are not fully understood and require further clarification.

The type I interferon response plays an essential role in antiviral defense by restricting the replication of the virus in infected cells [4]. Conversely, the sustained release of type I interferon switches on anti-inflammatory pathways, including the production of interleukin-10 (IL-10) and the expression of programmed cell death ligand 1 (PD-L1) [5]. Thus, IL-10 and PD-L1 exert inhibitory functions on virus-specific CD8^+^ T cells and account for subsequent exhaustion of CD8^+^ T cells [5]. In contrast, during viral infection, PD-L1 expression arises in infected cells and allows to avoid deterioration by cytotoxic T lymphocytes [6]. Consequently, upregulation of PD-L1 on target cells with concomitant exhaustion of CD8^+^ T cells may protect at-risk patients from serious virus-triggered immunopathology.

Ubiquitination is an important mechanism of posttranslational modification of proteins; it plays an essential role in the regulation of protein activity and cell homeostasis [7]. Deubiquitinases, such as cysteine proteases, remove ubiquitin moieties from several substrates, thereby leading to protein activation or inactivation, DNA repair, gene regulation, signal transduction, and degradation of proteins [7,8,9]. In the past decade, more than 100 deubiquitinase enzymes have been detected in humans. These can be divided into six different families. The largest subfamily includes the ubiquitin-specific peptidases (USPs), which account for more than 50% of all deubiquitinases [9,10]. Members of the USP group are highly conserved and are composed of three subdomains resembling the right finger, the thumb, and the palm [9,11].

Ubiquitin-specific peptidase 22 (Usp22) belongs to the USP family and is an integral component of the deubiquitinating (DUB) module of the Spt-Ada-Gcn5 acetyltransferase (SAGA) complex [6,7,8]. As part of this complex, Usp22 as a part of SAGA complex is able to deubiquitinate ubiquitinated histones and nonhistone proteins, including transcription factors [7,9,12,13]. The enzymatically active Usp22 within the DUB molecule of the SAGA complex mainly catalyzes the deubiquitination of the ubiquitinated histones H2A and H2B; this action results in the regulation of gene promotors and intergenetic regions and subsequently alters the transcription of diverse downstream genes [13,14,15,16]. Thus, Usp22 is involved in controlling a variety of cellular functions. On the other hand, in regulating cellular processes, Usp22 interacts with a number of nonhistone substrates, such as TRF1, SIRT1, NFAT, or cyclin B1, thereby initiating post-transcriptional regulation of these nonhistone proteins [7,9,12]. Usp22 participates in the regulation of the cell cycle and in the processes of metabolism, cell development, and apoptosis [7,17,18,19,20]. The function of Usp22 has been investigated in multiple cancer cell lines; these studies showed that the activation of Usp22 due to its overexpression tended to promote cell survival pathways and inhibit apoptosis [7]. With respect to these pro-mitotic characteristics, which are directed at cell survival and cell cycle progression, Usp22 is considered a potential oncogene [21]. However, in murine cancer models, Usp22 exerted both pro-oncogenic and anti-oncogenic effects [21]. More recent studies have shown a correlation between the overexpression of Usp22 and poor survival rates, the occurrence of metastases, and resistance to therapy with frequent recurrence of various types of cancer [7,21,22,23,24,25,26].

Recently, evidence of regulatory functions of Usp22 in innate immune system responses has increased [27,28,29,30]. Usp22 has been linked to the regulation of the interferon response by the deubiquitination of histone H2B, which serves to activate interferon signaling pathways [27]. Therefore, the loss of Usp22 in the hematopoietic system generates increased levels of locus-specific H2Bub1, followed by up-regulation of many interferon-stimulated genes in hematopoietic cells. Interferon signaling and downstream pathways induced by interferon-stimulated genes are crucial for antiviral immunity. While Usp22 can affect the expression of a large set of interferon-stimulated genes, it may be considered a potential key player in antiviral defense [28]. The results of previous in vitro studies support this presumption by demonstrating the negative regulator functions of Usp22 in type I and III interferon-mediated pathways after viral infection [28,29,30]. In contrast, Usp22 deficiency causes augmented activation of pathways involved in type I and III interferon signaling [27,28].

The lymphocytic choriomeningitis virus (LCMV) belongs to the Arenaviridae family [31]. LCMV is an enveloped, negative, single-stranded RNA virus [31]. The virus genome consists of two ambisense genomic RNAs (L and S) [32]. The small S segment codes for the glycoprotein precursor, the virion glycoproteins GP-1 and GP-2, and the virus nucleoprotein (NP) [33]. The large L segment encodes the small ring finger protein (Z-protein) and a high molecular mass protein (L-protein) serving as a viral RNA polymerase [33]. LCMV exerts no cytolytic activity; it infects cells by the interaction of its GP1 protein with the alpha-dystroglycan receptor located on diverse cell types [34]. LCMV is a common rodent pathogen [35]. In mice, LCMV can cause viral meningoencephalitis and severe hepatitis [34,35,36]. LCMV infection may be principally transmitted to humans by rodents [35]. Human LCMV infection frequently takes an asymptomatic course but occasionally causes flu-like symptoms or acute aseptic meningitis [35].

As a prototypic viral model, LCMV has been widely used in immunological studies to establish the key concepts of viral immunopathology [35]. Numerous strains of LCMV have been described [35]. Old strains, such as the Armstrong strain or the WE strain, discovered at the beginning of the 1930s, lead to acute LCMV infection with a robust cytotoxic lymphocyte response in adult mice; the virus is eliminated quickly, usually within approximately one week [37,38,39]. The cytotoxic T cells recognize viral antigens presented by the major histocompatibility complex I; these antigens induce the perforin-mediated killing of the target infected cells [37,38]. The CD8^+^ T cell response after acute LCMV infection is restricted to two immunodominant peptides (GP33-41 and NP 396-404) because of the high affinity of the cytotoxic T cells to these LCMV epitopes [37]. During the first three days after acute LCMV infection, plasmacytoid dendritic cells produce interferon type I [39]. The upregulation of the expression of costimulatory markers by dendritic cells is directed by interferon type I [39]. Interferon type I signaling also contributes to T cell activation, which maximizes the proliferation of CD8^+^ T cells [39]. Dendritic cells are also potent stimulators of CD8^+^ T cells after LCMV infection [39]. The expression of inhibitory receptors PD-L1 and PD-L2 on antigen-presenting cells increases after acute LCMV infection [39]. Moreover, natural killer cells are activated in an interferon type I-dependent manner [35,36,37,38,39]. The fact that the response of cytotoxic CD8^+^ T cells reaches its peak response on day 7 after infection is mainly responsible for LCMV clearance, whereas only low levels of neutralizing antibodies are detected in response to murine LCMV infection [38,39].

Although the interferon response is necessary for the control of LCMV infection, and Usp22, in turn, is involved in the interferon type I response, our study was aimed at analyzing how conditional knockout of Usp22 in the hematopoietic system affects the course of acute LCMV infection. LCMV infection of Usp22 deficient mice was associated with a decrease in PD-L1 expression on antigen-presenting cells and resulted in overactivation of CD8^+^ T cells. The accelerated function of CD8^+^ T cells leads to severe immunopathology accompanied by the recruitment of neutrophil granulocytes and monocytes in the infected liver.

## 2. Materials and Methods

### 2.1. Mice

The Usp22 KO (Usp22^fl/fl^ Vav1-Cre mice) were generated by Dietlein et al. as previously described and had a pan-hematopoietic deletion of the Usp22 protein [27]. The Usp22 were provided by Nikolaus Dietlein from the Division of Cellular Immunology, German Cancer Research Center, Heidelberg, Germany. The mice were housed in individual, ventilated cages under specific pathogen-free conditions. During survival experiments, the health status of the mice was checked twice daily. All animal experiments were authorized by the Landesamt für Natur, Umwelt und Verbraucherschutz Nordrhein-Westfalen (81-02.04.2019.A143 LANUV, Recklinghausen, Germany) and were performed in compliance with the German law for animal protection (TierSchG). Animals exhibiting severe symptoms of sickness or showing substantial weight loss during infection were put to death and considered dead for statistical analysis.

The mice with a lack of Cre recombinase or only one floxed site displayed wild-type phenotype and were considered as control animals.

### 2.2. Virus and Plaque Assay

LCMV strain WE was originally obtained from F. Lehmann-Grube (Heinrich Pette Institute, Hamburg, Germany) and propagated in L929 cells as previously described [40]. In all experiments, the LCMV WE was injected intravenously at the dosage of 2 × 10^5^ PFU per mouse. Viral titers were quantified in a plaque-forming assay on MC57 fibroblasts as previously described [41]. In brief, organs were harvested in a DMEM medium containing 2% FCS and homogenized using a TissueLyser (Qiagen, Hilden, Germany). Diluted virus samples were added to the L929 cell culture on the 24-well plate. After 3 h, the plate was covered with methylcellulose 1% medium. After 2 days of incubation, viruses were stained against LCMV nucleoprotein via an anti-LCMV-NP antibody (clone VL-4, Bio X Cell, Lebanon, PA, USA).

### 2.3. Cell Depletion

Depletion of CD8^+^ T cells was performed with an anti-CD8 antibody (anti-mouse CD8a antibody, clone YTS 169.4, Bio X Cell, Lebanon, PA, USA). The anti-CD8 antibody was applied intraperitoneally at the dosage of 100 ug per mouse one day before LCMV infection, on the day of infection, and then every second day. Monocytes and neutrophil granulocytes were depleted using the anti-Gr-1 antibody (anti-mouse Ly6G/Ly6C antibody, clone RB6-8C5, Bio X Cell, Lebanon, PA, USA), which was injected intraperitoneally at a dosage of 500 ug per mouse on day −1 before LCMV infection, on the day of infection, and then every second day.

### 2.4. ALAT, ASAT, and LDH Measurement

The activity of liver enzymes including alanine amino transferase (ALAT) and aspartate amino transferase (ASAT), as well as the concentration of lactate dehydrogenase (LDH), were measured in murine serum in the German Laboratory of the University Hospital Essen. The quantification of transaminases and LDH was required to assess the liver damage due to the acute murine LCMV infection primarily attacking the liver.

### 2.5. Histology

For immunofluorescence assay, snap-frozen tissue sections were stained with mouse monoclonal antibodies to CD11b (clone M1/70), CD11c (clone N418), Ly6C (clone HK1.4), Ly6G (clone 1A8-Ly6G), CD8 (clone 53-6.7), F4/80 (clone BM8), IgM (clone 11/41), PD-L1 (clone MIH5) (Thermo Fischer Scientific, Waltham, MA, USA), and CD169 (clone MCA884F, Bio-Rad Laboratories, Hercules, CA, USA). Histologic analysis of snap-frozen liver sections for detection of LCMV particles was performed with mouse monoclonal antibody to LCMV nucleoprotein (NP, made in-house). Images were acquired with the ZEISS Axio Observer Z1 (Carl Zeiss AG, Oberkochen, Germany).

### 2.6. Flow Cytometry

Experiments were performed using a FACS Fortessa (BD, Franklin Lakes, NJ, USA) and analyzed via FlowJo software (Ashland, OR, USA). For surface molecule staining, single suspended cells were incubated with antibodies for 20 min at 4 °C.

For the tetramer staining, 20 µL blood was incubated with allophycocyanin (APC)-labeled gp33 MHC class I tetramers (gp33/H-2Dd) for 15 min at 37 °C [42]. This LCMV-gp33 tetramer was provided by the Tetramer Facility of National Institute of Health (NIH). After incubation, the samples were stained with the anti-CD8 antibody (PECy7-conjugated, clone 53–67, Invitrogen, Thermo Fisher Scientific, Waltham, MA, USA) for 30 min at 4 °C. Erythrocytes were then lysed using 1 mL BD lysing solution (BD Biosciences), washed 1×, and analyzed with a flow cytometer. Absolut numbers of gp33-specific CD8^+^ T cells/μL blood were calculated from FACS analysis using fluorescing beads (BD Bioscience, Franklin Lakes, NJ, USA).

For measurement of intracellular interferon-γ production, single suspendered cells were stimulated with LCMV-specific peptides gp33 for 1 h [42]. Brefeldin A (Thermo Fischer Scientific, Waltham, MA, USA) was added for another 16 h incubation at 37 °C followed by staining with anti-CD8 (Thermo Fischer Scientific, Waltham, MA, USA). After surface staining, cells were fixed with 2% formaldehyde in PBS for 20 min, followed by permeabilization with 1% saponin in FACS buffer at room temperature and stained with anti-interferon-γ antibody for 30 min at room temperature (clone XMG1.2; eBioscences, Thermo Fisher Scientific, Waltham, MA, USA).

### 2.7. Multiplex Assay

Serum cytokines were analyzed with LEGENDplex™ (BioLegend, San Diego, CA, USA) according to the instructions and recommendations of the manufacturer. Serum cytokines were analyzed with LEGENDplex™ Mouse Anti-Virus Response Panel (13-plex) (BioLegend, San Diego, CA, USA); LEGENDplex is a multiplex immunoassay based on fluorescence-encoded beads and flow cytometry measurements. Briefly, before the assay, all samples were thawed at room temperature and centrifuged for 5 min at 1000× *g*. The samples were then pre-diluted 1:2 and incubated for 2 h with monoclonal capture antibody-coated beads. Following this, the beads were washed and incubated for one hour with biotin-labeled detection antibodies. Finally, the samples were incubated with streptavidin-PE for 30 min. After staining, beads were acquired by FACS Fortessa flow cytometry (BD Bioscience, Franklin Lakes, NJ, USA) with BD FACSDiva™ software (BD Bioscience, Franklin Lakes, NJ, USA). To determine cytokine concentrations for each sample, we used the LEGENDplex Data Analysis Software (BioLegend, San Diego, CA, USA) and followed the manufacturer’s protocol to extrapolate the results from standard curves.

### 2.8. Statistical Analysis

To detect statistically significant differences between two groups Student’s *t*-test was used. For analysis of differences in survival rates, the log-rank test was carried out. All mentioned experiments were conducted at least twice with similar results. Data are expressed as mean ± S.E.M. The level of statistical significance was set at *p* < 0.05. All data analyses were calculated with GraphPad Prism version 6 (GraphPad Software, Inc., La Jolla, CA, USA).

## 3. Results

### 3.1. Loss of Usp22 Results in Diminished Production of Type I Interferon at the Early Stage of Acute LCMV Infection without Impairment of LCMV Clearance

Usp22 has been reported to affect the expression of type I interferon genes and interferon-stimulated genes in the absence of infection or inflammation [27]. To understand the effect of Usp22 on antiviral immune responses, we challenged Usp22 deficient and wild-type (WT) mice with 2 × 10^5^ PFU LCMV WE strain. First, serum levels of type I and II interferons were determined during the course of acute LCMV infection. At the early stage, day 1 after infection, serum concentrations of interferon alpha and beta were lower in Usp22-deficient mice than in WT mice (Figure 1A). Serum levels of interferon-gamma had decreased at day 4 after LCMV infection in Usp22 deficient mice (Figure 1A). However, on day 9 after LCMV infection, concentrations of interferon beta were significantly higher in Usp22 deficient mice than in control mice (Figure 1A). In the naïve state and on day 5 after LCMV infection, serum levels of type I and II interferons were similar between WT and Usp22 deficient mice (Figure 1A).

We speculated that the reduction in interferon production caused by the loss of Usp22 may result in impaired clearance of LCMV and provoke the persistence of LCMV. There was no statistically significant difference between the Usp22 deficient mice and the control mice in virus titers in the spleen obtained at the early stage (day 2) of LCMV infection (Figure 1B). No relevant alterations in LCMV titers were seen in the liver on day 3 after infection in either KO or WT animals. Consequently, the Usp22 deficiency-related decrease in type I interferon synthesis immediately after LCMV infection led to a slight impairment of virus clearance in the spleen at the early stage of LCMV infection but did not significantly influence LCMV replication in the long term. Surprisingly, no relevant differences in viral load were observed in the liver, spleen, lung, and kidney on day 9 after infection (Figure 1C). Although the early interferon response in the first days after acute LCMV infection was reduced in Usp22 deficient mice, the virus titers were comparable in Usp22 deficient and wild-type mice after infection, a finding indicating that LCMV clearance is not significantly influenced by Usp22 knockout.

### 3.2. Downregulation of PD-L1 Expression on Antigen-Presenting Cells after LCMV Infection in Usp22-Deficient Mice

As previously reported, a Usp22-dependent mechanism regulates the expression of PD-L1 on antigen-presenting cells and tumor cells [21,43]. To address the influence of Usp22 on PD-L1 during acute LCMV infection, we first tested PD-L1 expression on CD11b^+^Ly6C^+^ Ly6G^−^ monocytes from peripheral blood collected on day 3 after active systemic LCMV infection. The results showed dramatically lower rates of PD-L1 expression on CD11b^+^Ly6C^+^ Ly6G^−^ cells in Usp22 deficient mice than in control mice (Figure 2A,B). Furthermore, we evaluated PD-L1 expression on CD11b^+^Ly6C^+^ Ly6G^−^ cells harvested from the spleen in Usp22 KO or wild-type mice on day 9 after infection. As shown in Figure 2A,B, the frequency of PD-L1 positive monocytes (Figure 2A) and the expression of PD-L1 on positive monocytes (Figure 2B) in the spleen after LCMV infection were significantly lower in Usp22 KO mice than in wild-type mice. The effect of Usp22 deficiency on PD-L1 expression in monocytes was more prominent in the spleen during the later course of LCMV infection than in blood tested at an early time point after acute infection. Visualization of PD-L1 expression in the spleen and liver tissue obtained on day 3 after LCMV infection also showed lower expression of PD-L1 in Usp22 deficient mice than in control mice (Figure 2C,D). The findings indicate the downregulation of PD-L1 on antigen-presenting cells in Usp22 deficient mice after systemic infection with LCMV.

### 3.3. Activation of CD8^+^ T Cell Immunity Promotes Immunopathology in the Liver of Usp22 Deficient Mice after LCMV Infection

The observed decrease in PD-L1 expression on antigen-presenting cells, which reflects the loss of inhibitory effects of antigen-presenting cells, may interfere with CD8^+^ T cells, leading to enforced activation of virus-specific CD8^+^ T cells. Interestingly, the LCMV-specific CD8^+^ T cell response against LCMV-gp33 was enhanced in Usp22 deficient mice (Figure 3A). In the samples obtained from the liver and spleen on day 9 after infection, the frequencies of CD8^+^ T cells were similar between Usp22 KO mice and control mice (Figure 3B). To gain additional insights into the function of CD8^+^ T cells, we stimulated liver and spleen tissue from infected animals with peptides derived from LCMV. The numbers of interferon-gamma-producing CD8^+^ T cells after re-stimulation of the liver cells were higher in Usp22 deficient mice than in control mice (Figure 3C). Interferon-gamma production by splenic CD8^+^ T cells in response to LCMV epitopes was similar between KO and WT mice (Figure 3C).

Next, histologic examination of the liver on day 9 after LCMV infection found infiltrates of neutrophil granulocytes and monocytes, as well as deposits consisting of IgM, whereas histologic analysis on day 3 after infection found no differences between Usp22 deficient mice and wild-type mice (Figure 4A). We also noted strongly diminished numbers of F4/80^+^ macrophages in particular on day 9 after infection in Usp22 deficient mice (Figure 4A). Immunofluorescence analysis showed colocalization of infiltrates consisting of granulocytes and monocytes with IgM deposits, as well as LCMV (Appendix A). Regarding histologic staining of the spleen tissue obtained on day 9 after LCMV infection for monocytes, granulocytes, B cells, and T cells, we did not recognize any differences between Usp22 deficient mice and wild-type mice. Spleen sections from Usp22 KO mice exhibited a limited expression of CD169^+^ macrophages, whereas sections from control animals did not (Figure 4B).

Usp22 KO mice exhibited a strong elevation of serum liver enzyme activity and lactate dehydrogenase activity during the follow-up period of acute LCMV infection, starting from day 7 after infection. This finding suggests the development of acute liver pathology related to LCMV infection (Figure 4C). Usp22-deficient mice also developed rapid weight loss and died after LCMV infection, whereas wild-type mice did not (Figure 4D).

Collectively, we hypothesized that the exaggerated CD8^+^ T cell response in Usp22 deficient mice infected with LCMV may be responsible for the accumulation of granulocytes and monocytes in the liver. This circumstance contributed to liver damage with lethal outcomes after acute LCMV infection.

### 3.4. Depletion of CD8^+^ T Cells Rescues Usp22 Deficient Mice from Liver Failure after LCMV Infection

Considering the enhanced response of CD8^+^ T cells in Usp22 KO mice after acute LCMV infection, we wondered whether depletion of CD8^+^ T cells might revert the detected liver immunopathology. For this purpose, we used a CD8^+^ T cell-depleting antibody that was administrated intraperitoneally before and after the systemic injection of the LCMV virus. Repeated flow cytometric measurement of CD8^+^ T cells from blood samples collected during treatment with monoclonal anti-CD8 antibody revealed sufficient depletion in treated Usp22 deficient and wild-type mice at all indicated time points (Appendix A). Infiltrates of neutrophil granulocytes and monocytes in the liver disappeared after treatment with anti-CD8 depletion antibody (Figure 5A). In addition, liver tissue did not stain for IgM after the depletion of CD8^+^ T cells (Figure 5A). Histologic analysis from liver tissue conducted on day 9 revealed increased LCMV load in Ups22 KO mice that were treated with the anti-CD8 antibody compared to those Usp22 KO mice that received PBS (Appendix A). Depletion of CD8^+^ T cells abrogated the increase in serum levels of liver enzymes and lactate dehydrogenase seen in Usp22 KO mice after LCMV infection (Figure 5B). On the contrary, the alternative application of anti-Gr-1 antibody depleting neutrophil granulocytes and monocytes only tempered but did not prevent the liver pathology caused by acute LCMV infection (Figure 5C). We found that the activity of transaminase and lactate dehydrogenase after administration of anti-Gr-1 antibody was slightly lower in Usp22 deficient mice with acute systemic LCMV infection than in control mice (Figure 5C). Thus, the onset of liver failure due to LCMV infection was delayed but not abrogated in Usp22 KO mice that received treatment with anti-Gr-1 antibody (Figure 5C). Consistently, when Usp22 KO mice were treated with anti-CD8 depletion antibody, survival rates of treated Usp22 KO mice were similar to those of WT animals after LCMV infection (Figure 5D).

Taken together, the results of CD8^+^ T cell depletion experiments suggest that stimulated CD8^+^ T cell immunity resulting from the absence of Usp22 may have caused severe hepatic immunopathology that was reversed by the administration of CD8^+^ T cell depleting antibody. However, liver pathology with subsequent liver failure was not affected by the use of the anti-Gr-1 antibody to target neutrophil granulocytes or monocytes forming the above-mentioned infiltrates.

## 4. Discussion

In the study reported here, we found that the absence of Usp22 in the hematopoietic system led to severe immunopathology in the liver following intravenous infection with LCMV. Usp22 deficient mice exhibited decreased levels of type I interferon on the first day after LCMV infection that did not relevantly limit virus clearance. The liver pathology was associated with liver failure occurring 7 days post-infection and was mediated by disturbed CD8^+^ T cell response leading to invasion of the liver by granulocytes and monocytes. The lack of Usp22 was associated with strong activation of LCMV-specific CD8^+^ T cells accompanied by augmented interferon-gamma production. Additionally, PD-L1 expression on monocytes was reduced in Usp22 deficient mice that underwent LCMV infection. Ablation of CD8^+^ T cells restricted immunopathology in the liver due to LCMV infection in Usp22 KO mice, whereas depletion of granulocytes and monocytes only prolonged the time until the appearance of acute liver failure. Collectively, these findings suggest that the enhanced immunity of virus-specific CD8^+^ T cells accompanied by the reduction of PD-L1 expression on antigen-presenting cells might have promoted liver immunopathology after acute viral infection with LCMV in Usp22 deficient animals.

Intriguingly, the research group of Dietlein et al. identified that the deficiency of Usp22 was related to the appearance of emergency hematopoiesis. Despite of absence of infection or inflammatory signals, mice lacking Usp22 in their hematopoietic system displayed overproduction of eeloid cells, in particular granulocytes, alteration in the development of B-cells, and enhancement in the expression of proinflammatory genes [27]. Loss of Usp22 exerted a protective effect in case of in vivo infection with Listeria monocytogenes [27]. Elevated proliferation of progenitors of myeloid cells accompanied by heightened phagocytosis capacity of neutrophil granulocytes was observed in infected mice having Usp22 deficiency [27].

First, we hypothesized that alterations in interferon release related to the absence of Usp22 might be critical for virus control and elimination. Lui et al. demonstrated a modulating function of Usp22 on type I interferon production and interferon-mediated pathways [28]. Increased Usp22 expression in vitro after corresponding transfection of 293T cells resulted in inhibited type I interferon formation upon infection with murine Sendai virus [28]. The infection of Vero cells overexpressing Usp22 with vesicular stomatitis virus (VSV) also led to reduced production of proinflammatory cytokines inclusive interferons, making the Vero cells more susceptible to VSV [28]. On the other hand, the replication of VSV, Sendai virus, and herpes simplex virus (HSV) was restrained after the knockdown of Usp22 in A549 cells [28]. Mechanistically, viral infection strengthens the cooperation of Usp22 with another deubiquitinase Usp13 that promotes cleavage of the stimulator of interferon genes protein (STRING) [28]. Moreover, Usp22 negatively influences the secretion of interferon-gamma [29]. On the contrary, the loss of Usp22 in human intestinal epithelial cells induced upregulation of diverse interferon-stimulated genes and interferon-gamma production in a STRING signaling-dependent manner and provided resistance against SARS-CoV-2 infection [29]. Therefore, we expected that Usp22 deficiency in our mouse model of acute viral infection might improve the viral clearance in relation to the enforced interferon production. However, we did not observe any relevant difference in virus control comparing Usp22 KO animals with controls. Our in vivo experiment data showed effects that were opposite to the in vitro data reported by Liu et al. [28]. We detected a diminishment of type I interferon levels on the first day after LCMV infection, and LCMV virus titers in the spleen were slightly elevated in Usp22 deficient mice, which is probably partly attributed to the decreased interferon production. In contrast to the previously reported in vitro data on the enhancement of interferon-gamma secretion due to the absence of Usp22 upon SARS-CoV-2-infection, we detected a decrease of interferon-gamma in Usp22 KO mice on day 4 after LCMV injection. Notably, data supporting the suppressive function of Usp22 on type I and II interferon production were derived only from in vitro experiments with infected murine or human cell lines. It is conceivable that, in vivo, numerous interactions of immune cells and cytokines might have provoked a decrease of interferon concentrations after LCMV infection under the conditions of Usp22 loss that could explain the differences in comparison to the previously published in vitro data. Dietlein et al. conducted experiments with mice having conditional knockout of Usp22 in the hematopoietic systems that were also used in our work [27]. Despite infection- or inflammation-free conditions, Usp22 KO mice displayed increased expression of inflammation genes, in particular interferon-stimulated genes, while systemic levels of interferon alpha and gamma remained similar between Usp22 KO mice and wild-types. However, our data on interferon are more in agreement with the observations of Cai et al. [29]. Besides the nuclear expression of Usp22, the colleagues identified a cytoplasmic synthesis of this molecule. Animals with a knockout or knockdown of Usp22 experienced a lethal course of infection with VSV or HSV-1 compared with respective controls [29]. Serum concentrations of interferon beta were significantly decreased in Usp22 deficient mice at 12 to 24 h post VSV or HSV-1 infection [29]. Thus, significantly increased virus titers of VSV and HSV-1 were detected on day 4 after infection [29]. To recover the corresponding pathomechanism for these findings, the authors proposed that cytoplasmic Usp22 removes ubiquitin conjugates from KPNA2, which serves for stabilization of KPNA2, which facilitates virus-triggered transmission of IRF3 into the nucleus and initiation of transcription of IRF3- dependent genes with antiviral properties [29]. Loss of cytoplasmic Usp22 and consequently reduced induction of IRF3-regulated downstream pathways upon LCMV infection might provide a possible explanation for the low interferon production found in Usp22 deficient mice in our study.

Nevertheless, in our study, no relevant changes in virus titers were detected at the late stage of infection when differentiating between KO and wild-type animals. Hence, we questioned whether the liver failure after acute LCMV infection with subsequent lethal outcomes in mice lacking Usp22 might be caused by the overwhelming immune response toward LCMV as the pathogen. In line with this hypothesis, we found strong infiltration of the liver tissue of infected Usp22 deficient mice by neutrophil granulocytes and monocytes. Dietlein et al. already described that the loss of Usp22 was associated with great potential toward immune enchantment [27]. The phenotype of the mice with hematopoietic Usp22 knockout, which was also used in the present study, was coined by emergency hematopoiesis with such hallmarks as the rise of myelopoiesis in the bone marrow and enhanced phagocytosis specific for only neutrophil granulocytes [27]. Increased phagocytosis capacity of neutrophil granulocytes was particularly apparent after bacterial infection of Usp22 deficient mice with Listeria monocytogenes, which might be considered to be an essential trigger [27]. Consistent with the data published by Dietlein et al., infiltrates of neutrophil granulocytes identified in the liver of LCMV-infected Usp22 KO animals might be a result of elevated myelopoiesis. In addition, based on the above-mentioned findings of Dietlein et al., severe deterioration in the liver upon LCMV infection might be explained by overactivated phagocytosis of neutrophil granulocytes inside the infiltrates. In light of our results, we also suggest that a viral infection with LCMV seems to be another potential trigger of granulocyte activation in the absence of Usp22.

Indeed, the depletion of granulocytes and monocytes was not efficient in restoring the proceeding liver damage after LCMV infection in Usp22 KO mice. Thus, we took into account other possible aspects that might lead to immunopathology. In fact, we saw activation of virus-specific CD8^+^ T cells in Usp22 deficient mice after administration of LCMV, which probably was promoted by the downregulation of PD-L1 expression on antigen-presenting cells. Recently, a large number of reports suggested Usp22 as a regulator of PD-L1 expression [21,43]. Usp22 stabilizes PD-L1 protein, protecting it against proteasome-mediated degradation in two ways: first via direct deubiquitination and second through interaction with the CSN5-PD-L1 axis modulating the CSN5 [21]. As a consequence, ablation of the Usp22 was shown to inhibit the PD-L1 release giving rise to the T cell-regulated cell killing [21,43]. In particular, in samples derived from solid cancers, aberrant expression of Usp22 was documented as providing T cell exhaustion and allowing evasion from the immune system, leading to cancer progression [21,22,23,24,25]. Further studies regarded Usp22 as a positive regulator of the transcription factor forkhead box protein 3 (FOXP3) of the regulatory CD4^+^ T cells [21,44,45]. Deletion of the Usp22 in regulatory T cells reduced their suppressive effects on cytotoxic CD8^+^ T cells [45]. That means, that the lack of Usp22 might be responsible for the disturbed function of regulatory T cells supporting the trends toward autoimmunity in Usp22 deficient mice. In this way, it is plausible, that immunopathology in the liver of LCMV-infected Usp22 deficient mice might correspond to the upregulation of CD8^+^ T cell-mediated immunity due to the loss of suppressive signaling directed by PD-L1. In the current study, the essential role of virus-specific CD8^+^ T cell activation in the development of liver failure emphasizes the results from the CD8^+^ T cell depletion experiments. There, the occurrence of liver failure after LCMV infection in Usp22 KO mice was successfully prevented by the treatment with a monoclonal anti-CD8^+^ T cell antibody. In light of our data, the appearance of infiltrates predominantly consisting of neutrophil granulocytes and monocytes is most likely a secondary phenomenon following enhanced CD8^+^ T cell response against LCMV. We also suppose the pronounced deposition of IgM colocalized with infiltrates of granulocytes and monocytes as a consequence of activation of granulocytes and monocytes involving stimulation of complement cascades.

We suppose an indirect effect of the Usp22 knockout on CD8^+^ T cells response mediated via downregulation of PD-L1 on antigen-presenting cells. Many reports indicated a reduction of PD-L1 release in the case of Usp22 knockout [21,43]. It was shown that PD-L1 expression increased in antigen-presenting cells in the first 72 h after acute LCMV infection with the Armstrong strain [39]. It is conceivable that the activation PD-L1-PD1 axis during the acute LCMV infection is required to limit the action of activated CD8^+^ T cells and avoid severe immune inflammation and immunopathology. When PD-1 knockout mice were infected with LCMV, they experienced an augmented T cells response that caused immunopathology [46]. The tissue damage was also partly due to the unlimited production of proinflammatory cytokines by activated T cells that attracted neutrophil granulocytes and macrophages in the tissue [46]. A similar scenario might have happened in our model of LCMV infection of Usp22 deficient mice provoking immunopathology in the liver.

In addition, the Usp22 deficiency in mice restricted to hematopoietic cells was found to correlate with a decrease in interferon type I [27]. Teijaro and colleagues reported that the suppression of the interferon type I signaling resulted in a significant inhibition of PD-L1 expression on antigen-presenting cells after LCMV infection [5]. Furthermore, the blockage of interferon type I led to increased numbers of CD8^+^ T cells and dendritic cells [5]. Hence, we speculate that in our model the downregulation of interferon type I due to the Usp22 knockout might have generated decreased PD-L1 expression on antigen-presenting cells that subsequently induced exaggerated and unlimited CD8^+^ T cell response with uncontrolled release of proinflammatory effector cytokines for the recruitment of neutrophils and macrophages promoting the destruction of the liver.

However, referring to our data on CD8^+^ T cell depletion with a monoclonal antibody in Usp22 deficient mice that revealed the loss of immunopathology in the liver and improved the survival of Usp22 deficient mice, we cannot fully exclude the direct effect of Usp22 knockout on CD8^+^ T cells. The CD8^+^ T cells are derived from the progenitor lymphocytes from the bone marrow. Thus, the Usp22 knockout might persist in CD8^+^ T cells. To clarify the question of whether Usp22 deficiency in the hematopoietic system might have an indirect or direct effect on CD8^+^ T cells, an isolated knockout of Ups22 only in CD8^+^ T cells is needed. Then, two different scenarios might be possible. In the case that the Usp22 knockout has a direct impact on the activation of CD8^+^ T cells, mice with Usp22 knockout in CD8^+^ T cells might die after acute LCMV infection due to the overactivation of CD8^+^ T cells and subsequent immunopathology. In contrast, if our presumption might be true and the immunopathology observed in the liver after acute LCMV infection is linked to an indirect effect of the Usp22 knockout on CD8^+^ T cells mediated by innate immune cells, in particular, antigen-presenting cells with downregulation of PD-L1 expression due to the Usp22 deficiency, mice having the Usp22 knockout only in CD8^+^ T cells might survive. An isolated Usp22 knockout in CD8^+^ T cells might rescue the mice from immunopathology and liver damage after acute LCMV infection because the Usp22 expression is not impaired in the antigen-presenting cells and, consequently, these cells would display regular PD-L1 expression, stopping the exaggerated T activation of CD8^+^ T cells during virus infection.

To answer the question of why the deletion of CD8^+^ T cells diminished the infiltration of neutrophils and monocytes in the liver but did not aggravate the course of LCMV infection, we developed the following considerations. Neutrophils and monocytes or macrophages are the key effectors of the cytotoxic CD8^+^ T cells dependent immunopathology, as it was previously demonstrated for the pathogenesis of acute meningitis after acute murine LCMV infection, causing vascular injury in murine central nervous system [35,47]. An increased CD8^+^ T cell activation promoted the expansion of neutrophil granulocytes and macrophages in the nerve tissue as a secondary phenomenon [35,47]. Similarly, in our research work, acute hepatitis in Usp22 deficient mice resulted from immunopathology directed by uncontrolled activation of CD8^+^ T cells following activation of myelomonocytic cells that initiated liver damage. Thus, it is highly likely that the loss of infiltrates containing the effector myelomonocytic cells was related to the depletion of CD8^+^ T cells in Usp22 deficient mice that as a result prevented liver damage and deterioration of the liver function.

Next, the usage of the monoclonal anti-CD8 antibody might have reversed the immunological response to LCMV infection in Usp22 deficient mice from immunopathology toward exhaustion of cytotoxic CD8^+^ T cells. According to prior studies, CD8^+^ T cell exhaustion was associated with an elevation of the viral load [48]. In concordance with observations from previous reports, we observed a higher LCMV load in Usp22 deficient mice with deletion of CD8^+^ T cells compared to those without (Appendix A). However, the LCMV has no cytolytic capacity; it is a cytopathic virus. Thereby, the animals suffering from CD8^+^ T cell exhaustion accompanied by increased LCMV load survived despite the chronic viral infection. The disturbed liver function and death of the Usp22 deficient mice occurred only due to the immunopathology after LCMV infection. The immunopathology was successfully limited by CD8^+^ T cell depletion in Usp22 deficient mice.

Nevertheless, we suggest that besides the strong CD8^+^ T cell response to LCMV antigens representing the best-studied mechanism in LCMV pathogenesis, there are other mechanisms that are involved in LCMV control. The most prominent mechanism is the interferon type I-dependent activation of natural killer cells that contributes to LCMV clearance as an unblocked alternative to the inhibited CD8^+^ T cell-mediated pathway [38]. Otherwise, inhibition of the interferon type I was reported to lead to elevated activation of CD4^+^ T cells that play a minor role in a regular course of acute LCMV infection [5]. Thereby, increased CD4^+^ T cell-mediated response might be another way to restrict the LCMV replication in the absence of adequate CD8^+^ T cell response. The lack of regulatory FOXP3 positive CD4^+^ T cells (Tregs) was described for the Usp22 deficient mice. In turn, the inhibition of Tregs was mentioned to be associated with intense B cell response and production of IgG [49]. Hence, we might also expect that the deficit of Tregs in Usp22 deficient mice might provide greater B cell response and antibody-mediated clearance of LCMV at a later time point.

In general, we reason, that the pathomechanism lying beyond the immunopathology detected upon LCMV infection of Usp22 deficient mice is multifactorial. Overall, our data support the conclusion that the downregulation of PD-L1 and enhanced functional CD8^+^ T cell response to LCMV might be one of the critical events for the development of liver damage after LCMV infection in the absence of Usp22. Further studies are required to fully address this point. In particular, in vivo experiments under conditions of specific Usp22 knockout in T cells might be helpful to better clarify the role of CD8^+^ T cell immunity in the context of LCMV infection. It is debatable whether the slight but significant decrease of interferon production at the early stage of acute LCMV infection leading to increased virus replication in the spleen might have contributed to the overactivated state of the innate immune system, especially of CD8^+^ T cells. The so-far-known, broadly preactivated interferon response due to the lack of Usp22 already in the absence of infection underlines the importance of Usp22 deficiency in the activation of the innate immune system [25]. The observations from our in vivo experiments in the Usp22 KO mouse model upon LCMV infection support the idea of the Usp22 absence as a stimulator of proinflammatory signaling and innate immune response, probably serving as a trigger of autoimmunity. Then, Usp22 seems to have negative regulatory effects in immunological processes, which might be beneficial for the coordination of overwhelming inflammation upon infections and controlling of autoimmune processes. Artificial induction of Usp22 or restoration of the deficient Usp22 production, probably as a result of genetic variations such as single-nucleotide polymorphism, might defend patients having, for example, COVID-19 or hepatitis C infection from an unfavorable course of infection marked by severe immunopathology and prevent the occurrence of autoimmune diseases.

Taken together, we showed the occurrence of severe liver immunopathology upon LCMV infection in mice lacking Usp22 in their hematopoietic system, which was mostly related to reduced expression of PD-L1 on antigen-presenting cells and overactivation of virus-specific CD8^+^ T cells resulting in recruitment of neutrophil granulocytes and monocytes infiltrating the liver and inducing acute liver failure.

## Figures and Tables

**Figure 1 vaccines-11-01563-f001:**
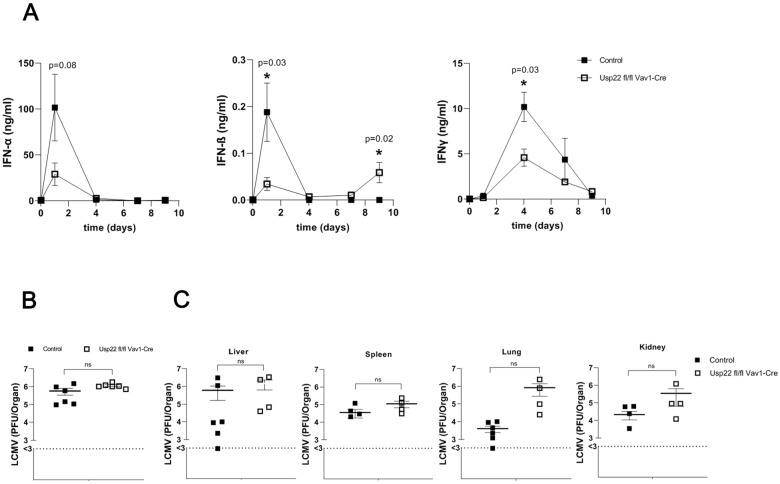
Usp22 deficiency is associated with the reduced production of type I and II interferons and with a slight increase in virus titers at the early stage of acute LCMV infection. (**A**–**C**) Usp22 deficient and WT mice were infected intravenously with LCMV WE (high dose, 2 × 10^5^ PFU per mouse) and examined at the indicated time points. (**A**) Serum levels of interferons α, β, and γ were measured in the serum by multiplex cytokine assay in Usp22 KO and WT mice on days 0, 1, 4, 7, and 9 after infection (n = 6). (**B**) LCMV titers from the spleen were measured by plaque assay on day 2 after infection (n = 6). (**C**) Viral titers were determined by plaque assay in the liver, spleen, lung, and kidney on day 9 after infection (n = 6). Data are shown as mean ± SEM and pooled from 2 to 3 independent experiments. *, *p* = 0.05; (**A**–**C**) unpaired two-tailed Student’s *t*-test.

**Figure 2 vaccines-11-01563-f002:**
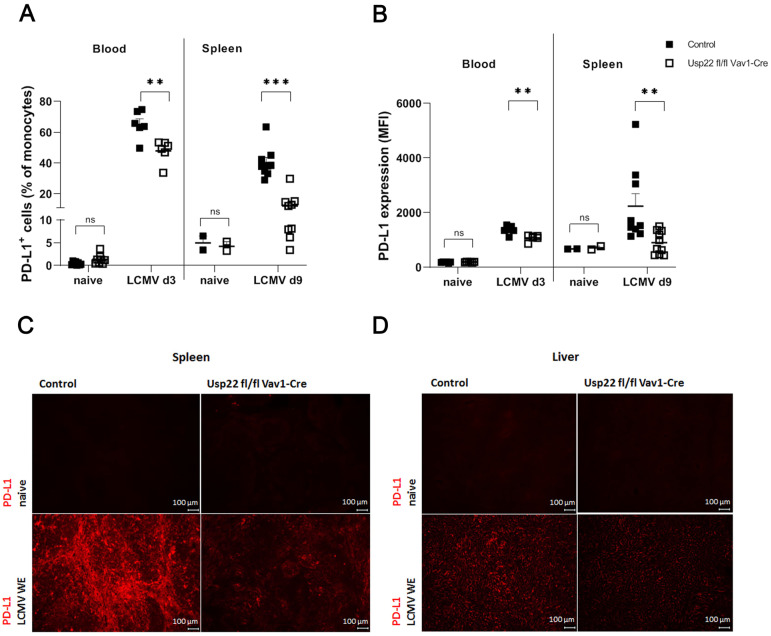
Conditional deletion of Usp22 in mice leads to downregulation of PD-L1 expression on antigen-presenting cells after acute LCMV infection. (**A**–**D**) We administered 2 × 10^5^ PFU of the LCMV WE intravenously to each mouse on day 0. (**A**,**B**) We measured PD-L1 expression on CD11b^+^Ly6C^+^ Ly6G^−^ monocytes. The percentages of CD11b^+^Ly6C^+^ Ly6G^−^ cells positive for PD-L1 (**A**) and the expression of PD-L1 on CD11b^+^Ly6C^+^ Ly6G^−^ cells (**B**) in the peripheral blood (n = 6) and spleen (n = 9) were determined by flow cytometry at the indicated time points. (**C**,**D**) Decreased expression of PD-L1 in the spleen (**C**) and liver (**D**) tissue of Usp22 KO and WT mice obtained on day 3 after infection and detected by immunofluorescence analysis. Scale bar = 100 μm; original magnification ×10; one representative of six is shown. Data are presented as mean ± SEM and are pooled from 2 to 3 independent experiments. **, *p* = 0.01; ***, *p* = 0.001; (**A**,**B**) unpaired two-tailed Student’s *t*-test.

**Figure 3 vaccines-11-01563-f003:**
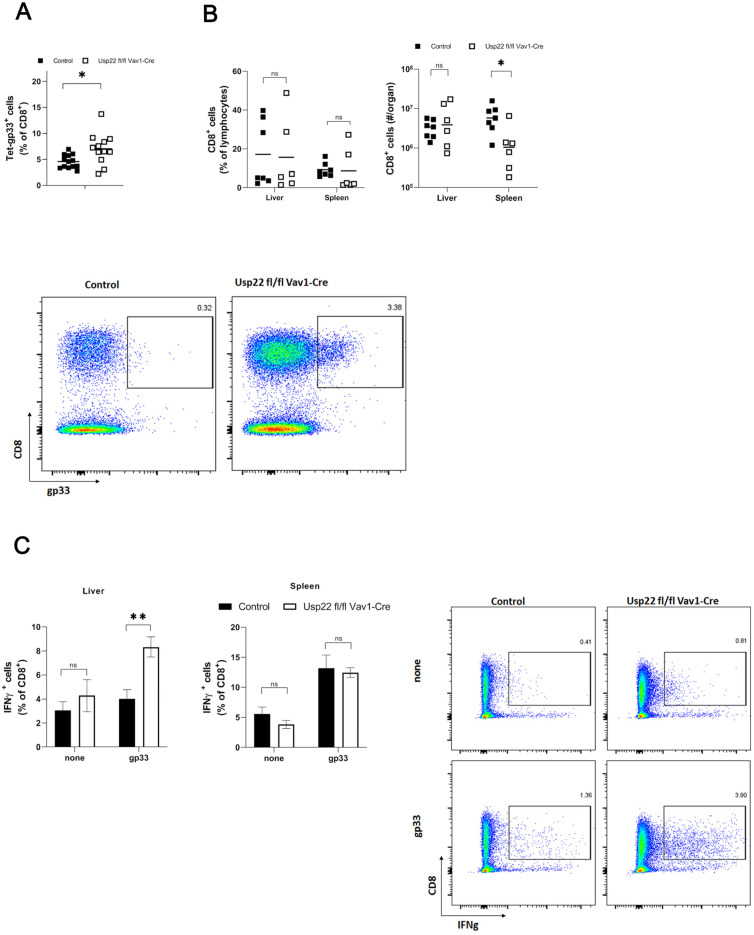
LCMV infection in Usp22-deficient mice results in an inflammatory response characterized by the activation of virus-specific CD8^+^ T cells. (**A**–**C**) Usp22-deficient mice and WT mice were infected intravenously with LCMV WE (2 × 10^5^ PFU per mouse) on day 0 and examined at the indicated time points. (**A**) LCMV-specific GP33 positive CD8^+^ T cells were assessed in the peripheral blood by tetramer staining and by flow cytometry on day 7 after infection (n = 10). (**B**) Numbers of CD8^+^ T cells in liver and spleen tissue obtained on day 9 after infection (n = 6) were determined by flow cytometry. (**C**) Intracellular IFN-γ production by CD8^+^ T cells derived from spleen and liver tissue of Usp22 KO mice and WT control mice (n = 6) was determined by flow cytometry after re-stimulation with LCMV specific peptides (gp33). Data from two or three independent experiments with consistent results are shown. *, *p* = 0.05; **, *p* = 0.01; (**A**–**C**) unpaired two-tailed Student’s *t*-test.

**Figure 4 vaccines-11-01563-f004:**
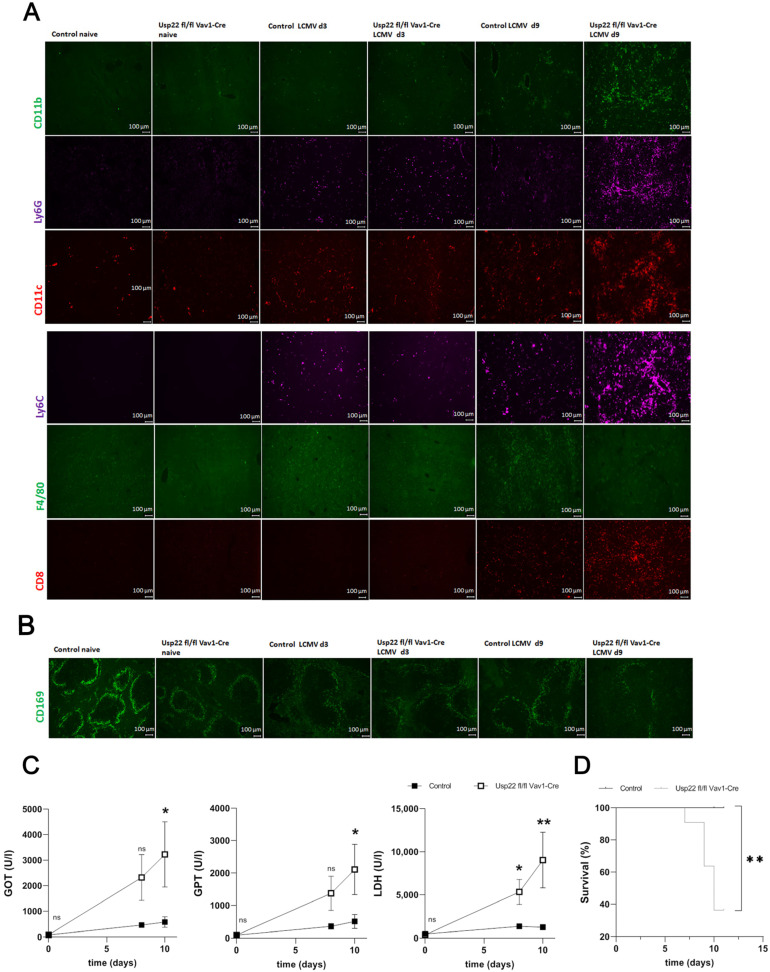
After acute LCMV infection, a deficiency in Usp22 leads to infiltration of the liver by monocytes and neutrophil granulocytes and to subsequent acute liver failure with a lethal outcome. (**A**) Immunofluorescence of liver sections from Usp22 KO and WT mice performed on days 3 and 9 after infection, stained for CD11b-, CD11c-, Ly6G-, and Ly6C-positive cells, CD8^+^ T cells, and F4/80+ macrophages. Scale bar = 100 μm; one representative of six is shown. Fluorescent microscopy images were captured at 10× magnification with a Keyence BZ-9000E microscope. (**B**) Three and nine days later, immunofluorescence analysis was conducted for CD169^+^ macrophages in snap-frozen spleen tissue from Usp22 KO and WT mice. Scale bar = 100 μm; one representative of six is shown. Original magnification ×10. (**C**,**D**) We injected 2 × 10^5^ PFU LCMV WE intravenously into each mouse on day 0. (**C**) Liver enzyme levels were monitored for the indicated period after LCMV infection (n = 11). (**D**) Survival analysis during acute LCMV infection, comparing Usp22 KO mice (n = 11) and WT mice (n = 11). Data are combined from 2 to 3 independent experiments with consistent results. *, *p* = 0.05; **, *p* = 0.01; (**C**) unpaired two-tailed Student’s *t*-test; (**D**) log-rank (Mantel–Cox).

**Figure 5 vaccines-11-01563-f005:**
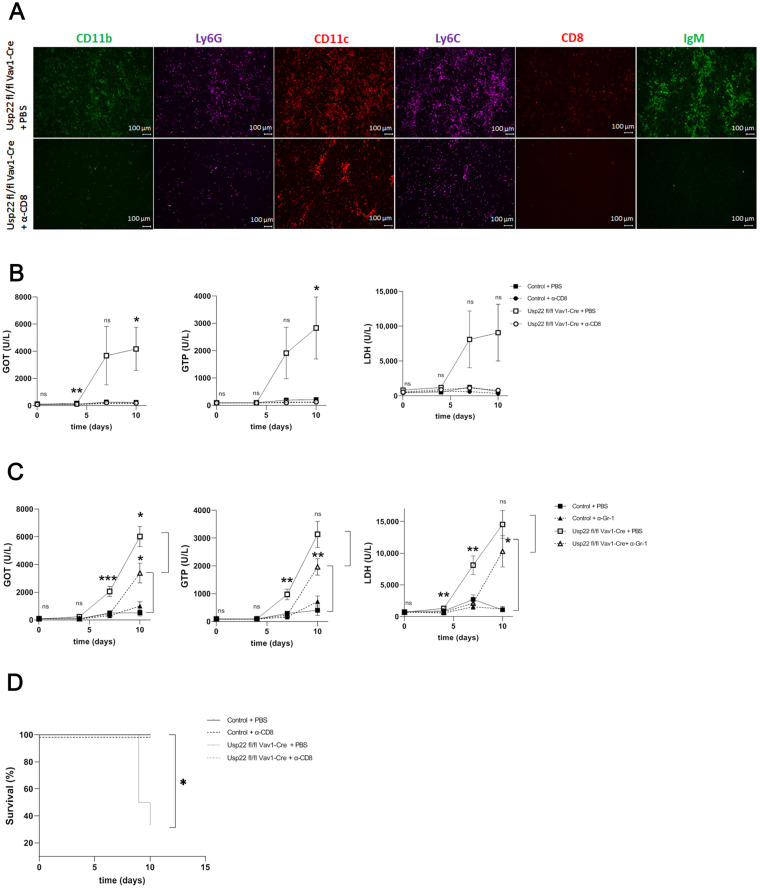
Although the depletion of CD8^+^ T cells rescues Usp22 deficient mice from acute immunopathology after LCMV infection, the depletion of neutrophil granulocytes and monocytes only delays the development of liver failure in Usp22 KO mice. (**A**–**D**) Before infection with LCMV WE (2 × 10^5^ PFU per mouse), Usp22 deficient mice and WT mice were treated intraperitoneally with 100 μg of the monoclonal anti-CD8a antibody (**A**) or with 500 μg of the anti-Gr-1 antibody (**C**) per mouse on day −1, day 0, and then every second day. PBS served as a control. Intravenous infection with LCMV WE (2 × 10^5^ PFU per mouse) was performed on day 0. (**A**) Immunofluorescence staining of snap-frozen liver sections was performed on day 9 after depletion of CD8^+^ T cells with subsequent infection. Scale bar = 100 μm; one representative of six is shown. Fluorescence microscopy images were captured at 10× magnification with a Keyence BZ-9000E microscope. (**B**,**C**) Liver enzyme levels were determined at the indicated time points after infection and before administration of monoclonal anti-CD8a antibody (**B**) or anti-Gr-1 antibody (**C**) (n = 6). (**D**) Survival rates of Usp22 KO and WT mice after LCMV infection and subsequent treatment with CD8^+^ T cell depleting antibody (n = 6). Results from 2 independent experiments are pooled. Data are shown as mean ± SEM. *, *p* = 0.05; **, *p* = 0.01; ***, *p* = 0.001; (**B**,**C**) unpaired two-tailed Student’s *t*-test; (**D**) log-rank (Mantel–Cox).

## Data Availability

The data presented in this study are available on request from the corresponding author. The data are not publicly available due to restrictions eg privacy or ethical.

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
