# Peer review of "Usp22 Deficiency Leads to Downregulation of PD-L1 and Pathological Activation of CD8+ T Cells and Causes Immunopathology in Response to Acute LCMV Infection"

_vaccines, 2023, doi:10.3390/vaccines11101563_

Round 1
Reviewer 1 Report
The current study used a conditional deletion of Usp22 in the hematopoietic and lymphatic systems to address its potential role in the immune response activated by acute LCMV infection. In wild-type mice, an acute virus infection induces type I interferon response and leads to restriction of virus replication. The authors describe a stepwise deregulated antiviral response where Usp22 deficiency impairs interferon response (partially), downregulates PD-L1 expression, and subsequently activates CD8+ T cells response that causes accumulation of granulocytes and monocytes in the liver leading to acute liver damage and death. The use of mouse genetics is a great advantage for this study and sheds light on several discrepancies in the role of Usp22 in the regulation of immune responses performed in vitro. My main concern is that the generic nature of Vav1-Cre leads to the deletion of Usp22 from all hematopoietic and lymphatic lineages, thus making the interpretation of the authors of this potential pathomechanism of Usp22 in antiviral response rather questionable. The discussion is carefully written and addresses several of these concerns. The results are not as well written however and this makes the logical arguments supporting the data hard to follow. If Usp22 is deleted in all hematopoietic lineages, could this deletion directly affect the function of CD8+ T cells? What would the authors predict to be the phenotype of mice with Usp22 deletion only in CD8+ T cells in their virus infection model? I have identified several points that need to authors attention and I am listing them below. I think after a careful restructuring of the introduction and the results sections and clarifications in the data and figures presented, this manuscript could be accepted for publication in Vaccines. Introduction: Line 55: "Deubiquitinases as cysteine proteases remove ubiquitin chains from several substrates leading to protein activation, inactivation, DNA repair, gene regulation, signal transduction and degradation of proteins [7-9]." USP dubs are cysteine proteases that remove ubiquitin moieties (single or multiple molecules in chains). Line 62: "Ubiquitin-Specific-peptidase 22 (Usp22) belongs to the USP family and builds up as a key component the Spt-Ada-Gen5 Acetyl transferase complex (SAGA) [6-8]." Builds up as a key component the SAGA complex doesn't make sense. The authors should write instead: and is an integral component of the de-ubiquitinating (DUB) module of the SAGA complex. Line 64-71: "The SAGA complex removes ubiquitin from the target transcription factors allowing transcription of downstream genes [12]". The DUB removes ubiquitin mainly from ubiquitinated histones H2B and H2A, and has been found to deubiquitinate several non-histone proteins including transcription factors (Atanassov et al., 2011). Usp22 has been shown to regulate a variety of cellular processes either through transcriptional regulation of gene promoters and intergenic regions through histone deubiquitination or through post-transcriptional regulation of multiple non-histone proteins. Line 72: "With regard to the ubiquitous expression of Usp22, its functions include regulation of cell cycle, metabolism, cell development, and apoptosis processes [7,16-19]. " Usp22 is not ubiquitously expressed in all the different organisms that it has been studied including drosophila, xenopus, and mice. Line 73-75: "In general, activation of Usp22 tends to promote cell survival pathways and inhibit apoptosis [7]. Due to these pro-mitotic characteristics which are directed to cell survival and cell cycle progression, Usp22 is considered as a potential oncogene [20]." The function of Usp22 has been tested in multiple cancer cell lines and overexpression of Usp22 is commonly linked with enhanced proliferation and reduced apoptosis. In mouse cancer models, however, Usp22 has been shown to have both pro-oncogenic and anti-oncogenic roles. Line 76: "Consequently recent studies reported the overexpression of Usp22 relying on poor survival, the occurrence of metastasis and recurrence of multiple cancer entities [7,20-24]." This sentence is wrong, several studies over the years have correlated Usp22 overexpression with poor survival, highly metastatic and therapy-resistant tumors (Glinskii et al.,2005). Line 92-95: the authors lack logical structure in their writing. In this paragraph, they should state their hypothesis in a single sentence, their approach and what was their main finding. Their statement control of LCMV infection is known to be strongly dependent on interferon response is not the main finding of this study. The introduction is lacking accuracy on several points and needs to be carefully revised. I think it would be more meaningful for the current manuscript to describe the pathogenesis of LCMV infection and how it leads to liver damage rather than listing multiple different viruses that are not relevant to this study or mentioning hepatitis C. Describing the molecular steps of the immune response after LCMV infection in mice would make it easier for the reader to understand the impact of Usp22 deletion in this same anti-viral response. Methods: Line 132: What are these liver enzymes and what was the purpose of their measurement? Line 164: The multiplex assay should be briefly described in the methods. Results: Line 178: "Usp22 was reported to affect the expression of type I interferon genes as well as interferon-stimulated genes in the absence of infection or inflammation. " what is the reference for this statement? Line 183: Indeed regarding interferon-gamma, serum levels declined at day 4 after LCMV infection in Usp22 deficient mice (Figure 1A). The authors describe interferon alpha and beta right before this sentence. "Indeed" does not make sense here. What do the authors conclude about the effect of Usp22 deletion on the viral load? Although the interferon response is not as robust, the viral load clearance is not different between wild-type and Usp22 deficient mice. Line 202: The authors should write "no statistically significant differences are observed". Also, they should add ns (not significant) in Figure 1C where they present these results. Line 212: "As previously described in the literature, Usp22 dependent mechanism regulate the expression of PD-L1 on antigen-presenting cells and tumor cells." What is the reference for this statement? Figure 2. "Deficiency of Usp22 is essential for downregulation of PD-L1 expression on antigen-presenting cells upon acute LCMV infection." The genetic deletion of Usp22 in mice by the authors cannot be essential for the downregulation of PD-L1. This sentence is not accurate scientifically. Either the function of Usp22 is essential for normal expression of PD-L1 or conditional deletion of Usp22 in mice leads to downregulation of PD-L1 expression. Line 227: it should be Figure 2 C and D and not Figure 4 C and D. Figure 2 C and D: there is no designation of the antibody used for staining on the figure and there are no scale bars. C should have a header to say spleen and D liver. Also, the images are very small and there is a lot of white space, the authors should increase the size of all the immunostaining images at least by 3x. Line 243: "The observed absence of PD-L1 expression on antigen-presenting cells..." A decrease in the expression of PD-L1 doesn't mean an absence of expression. The authors should rewrite their conclusions more accurately reflecting the actual data presented in the previous figure. Line 250: Why restimulation of the liver and spleen with LCMV peptides was necessary to observe the impact of Usp22 deletion on interferon gamma-producing T cells?? Why the impact was only observed in the liver? Figure 3: ns should be added when the comparison of wild type and Usp22 deficient cells do not show a statistically significant difference. Also, numbers of cell percentages should be added in all the flow cytometry quadrants. Line 266: Why the same histological examination is not performed in the spleen and the authors only stain the spleen for macrophages - specific markers? Figure 4: The images are too small, the size needs to be increased at least 3 times. The panel shown in the supplementary data is much better formatted. However, the magenta color used for Ly6G is not visible in panel A in the supplementary Fig.1. Also do the authors use markers for Ly6G and Ly6C in the Supp Fig1 A and B or is it a typo? Scale bars are still missing from all the microscope images. In Fig.4C there are no p-values on the graph. In Fig.4D the graph does not show n=11 for each genotype. A replotting of the survival data is necessary. The same applies to Fig.5D. Line 294: The authors state - Collectively, we supposed activation of CD8+ T cell response related to the absence of Usp22 to be responsible for the accumulation of granulocytes and monocytes in the liver contributing to the liver damage with lethal outcome after acute LCMV infection. This sentence needs restructuring.
The introduction and the results need thorough revision and restructuring, multiple conjunctions and conjunctive adverbs are misused and this makes the logical arguments built up by the experimental flow hard to follow. The discussion is much better structured and easy to understand.
Reviewer 2 Report
The authors used the conditional knock-out mice to clarify that the lack of Usp22 declined PD-L1 expression on APCs and consequently generated a pathological CD8+ T cell response, which gave rise to severe disease in the mice with acute infection of LCMV, as compared to the WT mice with the same infection. These results are interesting and data are sounded. However, the mechanism underlying the results remains unclear. So I have several concerns as below:
1. How dose the Usp22 affect PD-L1 expression onto activated APCs? Why not to affect the APCs at late stage of infection?
2. Why does the deletion of CD8+ T cells diminish the infiltration of N and M cells in organs, but not to lead the LCMV infection more severe?
3. After anti-CD8 antibody treatment, what are the percentages of CD8+ T cells in PBMC or spleen?
4. Why not to present the HE staining or confocal data of liver sections which can confirm the killing of activated CD8+ T cells against the LCMV-infected tissue cells?
Round 2
Reviewer 1 Report
I read carefully the responses to my concerns and all the edits and corrections in the second version of the manuscript. I am satisfied with the improvements and I only have 3 minor comments below : 1. If the authors agree that their model doesn't cover the potential indirect effects of Usp22 deficiency in different immune cells like CD8+ T cells, they should adjust their title accordingly. The current title sounds like the authors have identified a direct role of Usp22 deletion in the overactivation of T cells after attenuated viral response. To be more accurate they should state that "Usp22 deficiency attenuates antiviral response and leads to downregulation of PD-L1 and pathological activation of CD8+ T cells." 2. The authors responded: "We also conducted the same histologic staining in the spleen as in the liver at the same indicated time points. However, we did not recognize any differences between Usp22 deficient mice and wild-type mice. Hence, the data were not presented in the manuscript. As illustrated in Figure 4B of the manuscript, we only saw less CD169+ macrophages in the spleen of Usp22 deficient mice in comparison to the wild-type controls. " This statement should be added in the manuscript and either state - data not shown or add the data as supplementary information. 3. Finally in the second version of the manuscript I downloaded I could not find the improved figures. I would like to see the new figures before the paper is accepted for publication.
The language has been greatly improved throughout the manuscript.
Author Response
Point-by-point-reply for the Reviewer 1
Reviewer 1
Comments and Suggestions for Authors
I read carefully the responses to my concerns and all the edits and corrections in the second version of the manuscript. I am satisfied with the improvements and I only have 3 minor comments below:
- If the authors agree that their model doesn't cover the potential indirect effects of Usp22 deficiency in different immune cells like CD8+ T cells, they should adjust their title accordingly. The current title sounds like the authors have identified a direct role of Usp22 deletion in the overactivation of T cells after attenuated viral response. To be more accurate they should state that "Usp22 deficiency attenuates antiviral response and leads to downregulation of PD-L1 and pathological activation of CD8+ T cells."
Answer to Question 1:
We adapted the title of the manuscript as follow: “Usp22 deficiency leads to downregulation of PD-L1 and pathological activation of CD8+ T cells and causes immunopathology in response to acute LCMV infection.”
- The authors responded: "We also conducted the same histologic staining in the spleen as in the liver at the same indicated time points. However, we did not recognize any differences between Usp22 deficient mice and wild-type mice. Hence, the data were not presented in the manuscript. As illustrated in Figure 4B of the manuscript, we only saw less CD169+ macrophages in the spleen of Usp22 deficient mice in comparison to the wild-type controls. " This statement should be added in the manuscript and either state - data not shown or add the data as supplementary information.
Answer to Question 2:
As requested, we added the following statement to the Results section of the manuscript “Regarding histologic staining of the spleen tissue obtained on day 9 after LCMV infection for monocytes, granulocytes, B cells and T cells, we did not recognize any differences between Usp22 deficient mice and wild type mice (data not shown).”
- Finally in the second version of the manuscript I downloaded I could not find the improved figures. I would like to see the new figures before the paper is accepted for publication.
Answer to Question 3:
We downloaded the revised figures again as a zip file and additionally, we included the new figures directly into the manuscript.
Round 3
Reviewer 1 Report
I read the authors comments and found all their edits in the latest version of the manuscript. There are few minor points that need to be improved in the figures with the microscopy images (the scale bar number is very small to read and the fonts in Fig.4A and B and Fig. 5A are also too small), but I think the paper should be accepted for publication.
Some minor language problems observed.